# Two Nematicidal Compounds from *Lysinimonas* M4 against the Pine Wood Nematode, *Bursaphelenchus xylophilus*

**Yixiu Sun** [1], **Chao Wang** [1], **Guicai Du** [1], **Wenjun Deng** [1], **Hong Yang** [1], **Ronggui Li** [1], **Qian Xu** [2,*] and **Qunqun Guo** [1,*]

1 College of Life Sciences, Qingdao University, Qingdao 266071, China; sun1786218@163.com (Y.S.); wangchao6903199@163.com (C.W.); duguicai@qdu.edu.cn (G.D.); dengwenjun@qdu.edu.cn (W.D.); 13963939639@139.com (H.Y.); lrg@qdu.edu.cn (R.L.)
2 College of Agriculture and Biotechnology, Heze University, Heze 274000, China
* Correspondence: xq710301@163.com (Q.X.); gqunqun@163.com (Q.G.)

**Abstract:** A rich source of bioactive secondary metabolites from microorgannisms are widely used to control plant diseases in an eco-friendly way. To explore ideal candidates for prevention of pine wilt disease (PWD), a bacterial strain from rhizosphere of *Pinus thunbergii*, *Lysinimonas* M4, with nematicidal activity against pine wood nematode (PWN), *Bursaphelenchus xylophilus*, was isolated. Two nematicidal compounds were obtained from the culture of *Lysinimonas* M4 by silica gel chromatography based on bioactivity-guided fractionation and were subsequently identified as 2-coumaranone and cyclo-(Phe-Pro) by nuclear magnetic resonance (NMR) and mass spectrometry (MS). The 2-coumaranone and cyclo-(Phe-Pro) showed significant nematicidal activity against PWN, with $LC_{50}$ values at 24 h of 0.196 mM and 0.425 mM, respectively. Both compounds had significant inhibitory effects on egg hatching, feeding, and reproduction. The study on nematicidal mechanisms revealed that 2-coumaranone and cyclo-(Phe-Pro) caused the accumulation of reactive oxygen species (ROS) in nematodes, along with a notable decrease in CAT and POS activity and an increase in SOD activity in nematodes, which might contribute to the death of pine wood nematodes. Bioassay tests demonstrated that the two compounds could reduce the incidence of wilting in Japanese black pine seedlings. This research offers a new bacterial strain and two metabolites for biocontrol against PWN.

**Keywords:** *Bursaphelenchus xylophilus*; 2-coumaranone; cyclo-(Phe-Pro); reactive oxygen species (ROS); nematicidal activity



## 1. Introduction

Pine wilt disease (PWD) is an extremely destructive forest disease caused by the pine wood nematode (PWN), *Bursaphelenchus xylophilus* [1]. The disease was discovered in Jiangsu Province in 1982 and subsequently spread rapidly in China [2]. The State Forestry and Grassland Administration of China announced that PWD was found in 721 county-level administrative regions across the country in 2021. It caused a serious economic loss and a negative influence on the ecological environment [3].

Owing to the rapid spread of PWD, effective and reliable prevention and control measures for the disease became more and more urgent. Physical controls, such as cutting and burning the infected pine trees, cannot fundamentally prevent the spread of PWN [4]. Spraying synthetic insecticides against PWN vectors, such as *Monochamus alternatus* and trunk injection of nematicides, to directly control PWN as the common measures for PWD management might cause environmental pollution and disruptions of the host or other beneficial organisms [5,6]. Biocontrol based on microorganisms is considered as a promising strategy in the green management of plant diseases. Fungi and actinomycetes producing mycotoxins or antibiotics are reported to be effective in PWN control [7–10]. However, bacteria are far more commonly reported for biocontrol of PWN than fungi and actinomycetes due to their high diversity and rapid growth. Yu et al. [11] reported the

nematicidal effect of volatiles produced by *Pseudotalteromonas marina* strain H-42 and *Vibrio atlanticus* strain S-16 that were isolated from the seawater and bay scallops, respectively. Ponpandian et al. [12] screened two nematicidal bacteria from the endophytes of four Pinus species, and found that their metabolites had a significant inhibitory effect on egg hatching and development of PWN.

The plant growth-promoting rhizobacteria, a group of root-colonizing microorganisms beneficial to plants, provide promising and sustainable sources for the screening of biological control microbes against plant pathogens [13]. Plant rhizoshperic microbes can indirectly promote plant growth by production of toxins, biosurfactants, and lytic enzymes to inhibit plant pathogens and induce systemic resistance in host plants [14]. *Pseudomonas fluorescens* CHA0 and *Bacillus amyloliquefaciens* FZB42 were identified as antagonists of root-knot nematodes [15,16]. Moreover, Deng et al. [17] reported that the microbial diversity of the pine root system changed significantly when PWNs infected trees at early and late stages, and these changes were associated with the suppression effect of microbial community against PWNs.

In the present study, we screened a strain of rhizosphere bacteria from healthy Japanese black pine trees with nematicidal activity against PWN, and characterized the effective compounds in its culture, which provided reference to exploration of nematicidal agents against PWN.

## 2. Materials and Methods

### 2.1. Sample Preparation and Nematode Cultivation

Soil samples were collected from the roots of *Pinus thunbergii* on the campus of Qingdao University. PWNs were isolated from infected *P. thunbergii* by the Baermann funnel and morphologically confirmed [18]. The collected PWNs were washed 3 times with sterile water and cultured with *Botrytis cinerea* on potato dextrose agar (PDA) medium in dark at 25 °C for 1 week.

### 2.2. Isolation and Screening of Nematicidal Bacteria against PWN

An approximately 0.1 g soil sample was transferred to 49.9 mL sterile water and incubated at 30 °C for 30 min. The soil suspension was diluted serially with sterile water to obtain $10^{-1}$ to $10^{-6}$ dilutions [19]. 100 μL soil suspension from each dilution was plated on nutrient agar (NA) culture medium, and incubated at 30 °C for 48 h. Single colonies were picked from plates and sub-cultured until a pure culture with nematicidal activity against PWN was obtained.

The purified bacteria were fermented in a nutrient broth (NB) medium at 30 °C for 4 d. The culture was centrifuged at 10,000 rpm for 10 min and the supernatant was collected for subsequent experiments. The impregnation method was applied to screen the bacteria with nematicidal activity [20]. Next, 400 μL of the supernatant was mixed with 100 μL PWN suspension (containing approximately 100 nematodes) in sterile water and added into a well of 24-well plate, whereas control groups were carried out by mixing NB medium and PWN suspension. The plates were incubated at 25 °C for 72 h under dark conditions. Dead nematodes were observed and counted under a stereomicroscope (Motic BA200, Xiamen, China). Nematodes were considered dead if they were stiff and had no response to physical stimulus. Corrected mortality was calculated according to Schneider Orelli's formula [21]:

$$\text{corrected mortality (\%)} = [(\text{mortality in treatment} - \text{mortality in control})/ (100 - \text{mortality in control})] \times 100 \tag{1}$$

The experiment was carried out with three biological replicates.

Three strains with obvious nematicidal activity against PWN were further studied for their nematicidal activity by diluting their supernatants to 2.5, 5, and 10-fold, respectively [22]. The nematicidal activity of diluted suspensions was tested following the same procedures as described above.

### 2.3. Identification of Bacteria with Nematicidal Activity

The bacterial strain M4 with the highest nematicidal activity was identified. After culturing the M4 strain on a NA plate at 30 °C for 3 days. The strain was characterized by gram staining, spore staining, capsular staining, and morphological observation under a microscope (Motic SW350B, Xiamen, China). In addition, biochemical tests including starch hydrolysis test and oil hydrolysis were also used as an indicator for the identification of the M4 strain [23].

The M4 strain was cultivated in NA liquid medium at 30 °C for 18 h, then the genomic DNA was extracted using the TIANGEN Genomic DNA Kit (TIANGEN Biotech, Beijing, China). The 16S rDNA was amplified by PCR using the genomic DNA and primer pair 27F (5′-AGAGTTTGATCCTGGCTCAG-3′) and 1492R (5′-TACCTTGTTACGACTT-3′) and sequenced by Sangon Biotech Company (Qingdao, China). The 16S rDNA sequence underwent a BLAST search in the GenBank database of the National Center for Biotechnology Information (NCBI). The phylogenetic tree was constructed in MEGA7 using the neighbor-joining (N-J) method [24,25].

### 2.4. Isolation and Structural Determination of the Nematicidal Compounds

The culture of strain M4 (5 L) was extracted with an equal volume of ethyl acetate three times [12]. The ethyl acetate phase was combined and concentrated to dryness by rotary evaporation at 25 °C under vacuum, and 0.5 g of crude extracts was obtained. The nematicidal bioactivity of the crude extracts and the left aqueous phase was then bioassayed. Next, 5 μL of ethyl acetate extract (10 mg/mL, dissolved in DMSO) was mixed with 95 μL nematode suspension (containing approximately 100 nematodes) in a 96-well plate at 25 °C for 72 h under dark conditions. Additionally, 5 μL of ethyl acetate extract of pure NB medium (10 mg/mL, dissolved in DMSO), mixed with 95 μL of the same nematode suspension, was served as control. Dead nematodes were observed and counted under a stereomicroscope (Motic BA200, Xiamen, China).

The nematicidal components in the crude extracts were isolated by silica gel column chromatography (200 to 300 mesh) with petroleum/ethyl acetate (2:1, 2:3, 1:3, and 1:5 *v/v*), ethyl acetate/methanol (20:1 *v/v*), and five fractions (Fr1, Fr2, Fr3, Fr4, and Fr5) were collected [26]. Fr4 and Fr5, which showed higher nematicidal activity, were further purified by silica gel chromatography based on bioactivity-guided fractionation. Compound **1** in Fr.4 was recrystallized with petroleum/ethyl acetate (1:5, *v/v*) and compound **2** in Fr.5 was recrystallized with ethyl acetate.

The NMR spectra were obtained using NMR spectrometer (JNM-ECZ600R, JOEL, Tokyo, Japan) operating at 600 MHz for $^1$H NMR and 100 MHz for $^{13}$C NMR in chloroform-*d* (CDCl$_3$) with tetramethylsilane (TMS) as the internal standard. The electron impact mass spectra (EIMS) were determined using gas chromatography mass spectrometry (7890A-5975C, Agilent, Santa Clara, CA, USA).

### 2.5. Nematicidal Activity of the Nematicidal Compounds against PWN In Vitro

Nematicidal activities of the isolated compounds were assessed by determination of the LC$_{50}$ value for 24 h following the method as previously described [27]. Corrected mortalities were determined for each compound according to Equation (1), and the LC$_{50}$ values were calculated according to the Probi model, respectively.

### 2.6. Effect of the Nematicidal Compounds on Egg Hatching

Eggs were obtained according to the method previously described by Liu et al. [6]. Specifically, approximately 100 nematode eggs were transferred to a 48-well culture plate containing 2-coumaranone or cyclo-(Phe-Pro) of different concentrations (0.1 mM, 0.2 mM, and 0.5 mM in 5% DMSO), while 5% DMSO was used as a control group. Plates were

incubated at 25 °C and nematodes at the J2 stage were counted at 48 h. Three replicates were performed in this experiment. The hatching rate was calculated as follows:

$$\text{hatching rate (\%)} = [\text{juveniles}/(\text{eggs} + \text{juveniles}) \times 100] \qquad (2)$$

### 2.7. Influence of the Nematicidal Compounds on Nematodes Feeding and Population

Approximately 100 nematodes were treated with 0.1 mM 2-coumaranone or cyclo-(Pro-Phe) solution in 5% DMSO for 24 h, and 5% DMSO and sterile water were used as controls. PWNs were transferred to a PDA plate covered with *B. cinerea* and incubated at 25 °C for 7 d. The feeding of PWNs to *B. cinerea* was observed and photographed daily. Nematodes in each group were then collected, washed with distilled water, and quantified under a stereomicroscope (Motic BA200, Xiamen, China) [28].

### 2.8. Detection of Reactive Oxygen Species (ROS) and Antioxidant Enzymes in Nematicide-Treated PWD

ROS production was determined by the fluorescein 2,7-Dichlorodihydrofluorescein-diacetate (DCFH-DA) probe method [29]. Approximately 200 adult nematodes were pretreated with 0.2 mM 2-coumaranone and 0.2 mM cyclo-(Phe-Pro), respectively. They were incubated in 96-well plates for 24 h, and 5% DMSO was used in control wells. At the end of incubation, the PWNs were washed three times with 0.01 M PBS and treated with a probe from ROS assay kit (Nanjing Jiancheng Bioengineering Institute, Nanjing, China). Reactive oxygen hydrogen donor-induced PWNs served as positive control. Nematodes were collected, centrifuged at 10,000 rpm for 5 min, and washed three times with 0.01 M PBS. Then, the fluorescence intensity was detected at an excitation wavelength of 485 nm and a blocking wavelength of 528 nm under a fluorescence microscope (Olympus IX73, Tokyo, Japan).

The treated nematodes were thoroughly ground on ice, the extracts were centrifuged at 4 °C, 10,000 rpm for 10 min, and the supernatants were collected. The protein content was determined according to the Bradford method. The activities of superoxide dismutase (SOD), peroxidase (POD), and catalase (CAT) were measured using azalantetrazole reduction, guaiacol, and hydrogen peroxide, respectively [30,31]. The absorbance of each reaction was measured at wavelengths of 560 nm, 470 nm, and 240 nm, using a UV spectrophotometer (GENESY-50, Thermo Fisher Scientific, Waltham, MA, USA), respectively.

### 2.9. Nematicidal Activity of the Nematicidal Compounds against PWN In Vivo

Healthy Japanese black pine seeds were soaked with $HgCl_2$ for 10 min and rinsed with sterile water three times. Seeds were placed in Petri dishes containing sterilized filter paper, which was kept moist and incubated in the dark at 25 °C for 5 d. When the seeds germinated with roots about 1.5 cm long, they were transferred to nutrient soil on daylight exposure for 16 h/d cultivation for 30 d. Next, 5 μL of nematode suspension (50 PWNs/μL) was injected into pine seedlings by micro syringe [32]. On post-inoculation day 2 (PID 2), each seedling in the experimental groups was injected with 10 μL of 2-coumaranone (50 mM) and cyclo-(Phe-Pro) (50 mM), respectively. The same volume of 5% DMSO was injected into pine seedlings for the control group. In addition, the pine seedlings without inoculation of PWN were injected with 10 μL of nematicidal compounds (50 mM) to observe whether they would have an unexpected effect on the seedlings, and the same dose of sterile water was injected into another group of seedlings as the control.

### 2.10. Statistical Analysis

The means and standard deviations (SD) of the corrected mortality data were calculated in Microsoft Excel. One-way analyses of variance (ANOVA) with least significant differences (LSD) test ($p = 0.05$) and Student's *t*-tests ($p = 0.05$ and $0.01$) were performed to compare among treatment levels, and the lethal concentration ($LC_{50}$) values were calculated by probit analysis in SPSS version 25.0 (SPSS Inc., Chicago, IL, USA).

## 3. Results

### 3.1. Screening and Identification of Nematicidal Strains

A total of 40 bacterial strains were isolated from soil samples. The corrected mortality of PWNs was determined by exposing PWNs to a 1.25 times diluted cell-free culture for 72 h (Table 1). The strains with higher than 80% corrected mortality were considered as nematicidal candidates. The corrected mortalities of the cultures from BG-1, S-1-3, and M4 strains were 86.63%, 88.63%, and 91.30%, respectively, which were significantly higher than that of other strains. After further dilution for 2.5, 5, and 10 times, the culture of the M4 strain showed the strongest nematicidal effect among the three forementioned strains (Figure 1). Thus, the M4 strain was selected for subsequent characterization studies.

**Table 1.** Nematicidal activities of cell-free culture of different strains.

| Stains | Corrected Mortality (%) | Stains | Corrected Mortality (%) | Stains | Corrected Mortality (%) | Stains | Corrected Mortality (%) |
|---|---|---|---|---|---|---|---|
| NA-1 | 5.36 ± 1.52 c | BG-1 | 86.63 ± 1.53 o | S-1-1 | 19.4 ± 1.53 g | M1 | 8.70 ± 1.73 e |
| NA-2 | 49.83 ± 3.00 n | BG-2 | 7.02 ± 2.08 d | S-1-2 | 37.79 ± 3.00 l | M2 | 1.00 ± 1.52 a |
| NA-3 | 2.01 ± 2.08 a | BG-3 | 29.77 ± 3.61 i | S-1-3 | 88.63 ± 1.15 o | M3 | 4.01 ± 2.08 a |
| NA-4 | 5.02 ± 1.53 b | BG-4 | 16.38 ± 1.53 g | S-1-4 | 11.71 ± 1.73 f | M4 | 91.3 ± 1.53 o |
| NA-5 | 9.03 ± 1.53 e | BG-5 | 7.02 ± 0.58 d | S-1-5 | 5.69 ± 1.73 c | M5 | 6.69 ± 1.73 d |
| NA-6 | 5.36 ± 1.52 c | BG-6 | 16.38 ± 1.53 g | S-2-1 | 16.34 ± 2.08 g | M6 | 11.71 ± 1.73 f |
| NA-7 | 27.09 ± 2.52 i | BG-7 | 20.07 ± 1.15 g | S-2-2 | 5.35 ± 1.52 c | M7 | 19.4 ± 2.08 g |
| NA-8 | 0.34 ± 0.58 a | BG-8 | 1.00 ± 1.53 a | S-2-3 | 36.11 ± 1.15 k | M8 | 1.00 ± 1.52 a |
| NA-9 | 50.17 ± 2.52 n | BG-9 | 22.07 ± 3.06 h | S-2-4 | 40.80 ± 2.64 m | M9 | 32.78 ± 2.00 j |
| NA-10 | 30.77 ± 3.00 i | BG-10 | 37.45 ± 2.52 l | S-2-5 | 18.06 ± 2.31 g | M10 | 37.12 ± 2.51 l |
|  |  |  |  |  |  | Control | 0.33 ± 0.75 a |

The mean corrected mortality of nematodes was determined after exposure to 1.25-fold dilution of cell-free cultures for 72 h. Control was NA medium. Data were mean ± SD of three replicates. Means in the column followed by the same letter did not differ significantly ($p = 0.05$) according to the LSD test.

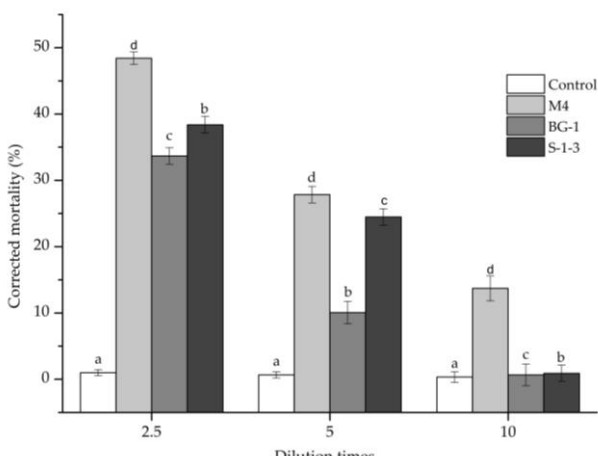

**Figure 1.** The nematicidal activity for the cell-free culture of strains M4, BG1, and S-1-3 diluted 10 times, 5 times, and 2.5 times, respectively. Data were given as mean. Means with the different letters were significantly different ($p < 0.05$) based on one-way ANOVA with multiple comparison analysis using the LSD test.

The colonies of the M4 strain were round with greenish-yellow color and entered margin. The cells were Gram-negative and motile with a short rod shape. Amylolysis, MR, and hydrogen sulfide tests were positive, while oil hydrolysis and Voges–Proskauer tests were negative. According to the comparison results of 16S rDNA sequences, the sequence from the M4 strain was homologous to that from *Lysinimonas* sp. F22 by 99.93%, which were both clustered in the same branch on the phylogenetic tree (Figure 2). Based on the morphological observation, analysis of physiological properties and 16S rDNA sequencing

comparation, the M4 strain belonged to the genus *Lysinimonas*, and therefore was named *Lysinimonas* sp. M4.

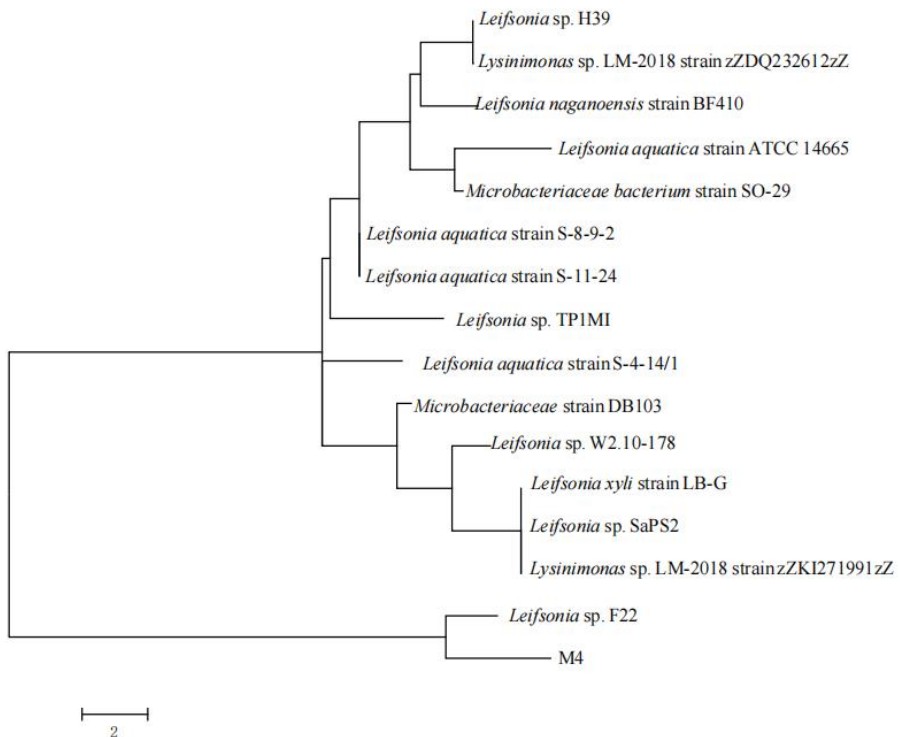

**Figure 2.** The phylogenetic tree constructed using a neighbor-joining method based on 16S rDNA gene sequence numbers at the nodes indicate the bootstrap values on neighbor-joining analysis.

### 3.2. Isolation and Identification of the Nematicidal Compounds from the M4 Strain

The nematicidal bioassay of ethyl acetate extract and the water phase of the *Lysinimonas* sp. M4 culture indicated that the ethyl acetate extract had significant nematicidal activity against PWN, while no obvious nematicidal activity was detected in the water phase (Table 2). Following a further isolation of ethyl acetate extract through silica gel column chromatography, five fractions (Fr. 1 to Fr. 5) were collected and tested for their nematicidal activity. The corrected mortality rate for Fr. 4 and Fr. 5 at a concentration of 200 µg/mL at 72 h was 91.87% and 70.68%, respectively, which was significantly higher than that for the other three fractions (Figure 3).

**Table 2.** The nematicidal activity of the aqueous phase and the ethyl acetate extract of the M4 strain culture for 72h.

| Smples | Mean Corrected Mortality ± SD (%) | | |
|---|---|---|---|
| | 24 h | 48 h | 72 h |
| Water phase | 1.00 ± 1.00 a | 2.00 ± 1.00 a | 2.00 ± 1.00 a |
| Ethyl acetate extract | 37.33 ± 2.309 b | 65.00 ± 2.00 c | 87.33 ± 2.08 d |

The mean corrected mortality of the nematodes was determined after exposure to the water phase and ethyl acetate extract at a different culture time. Data were mean ± SD of three replicates. Means in the column followed by the same letter did not differ significantly ($p$ = 0.05) according to the LSD test.

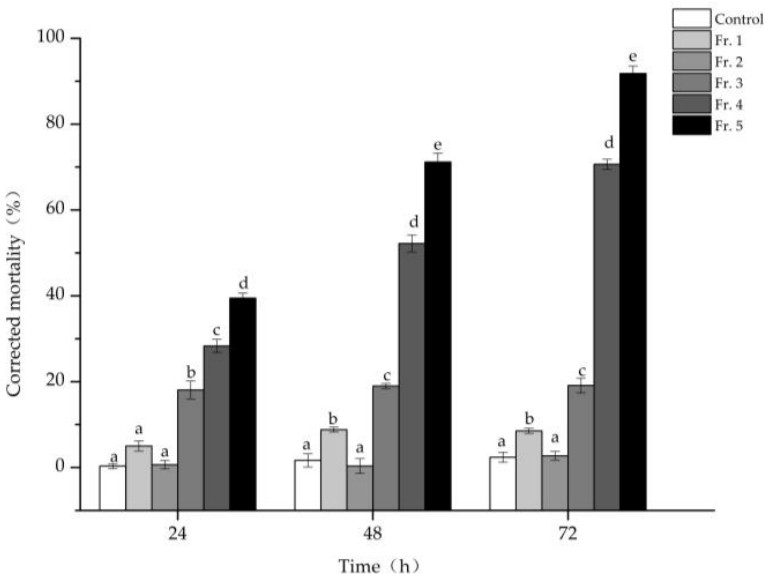

**Figure 3.** Corrected mortality of PWNs for different fractions. PWNs were treated with 200 μg/mL Fr. 1 - Fr. 5 effluent at 24 h, 48 h, and 72 h, respectively, and the corrected mortality was determined. Mean values were plotted with standard deviations (SD). The control group only contained NB medium. Means with the different letters were significantly different at $p < 0.05$ based on one-way ANOVA with multiple comparison analysis using the LSD test.

The nematicidal components in Fr.4 and Fr.5 were further purified and identified. The compound **1** derived from Fr. 4 was a pale-yellow crystal with a molecular weight of 134.037. Its molecular formula was determined to be $C_8H_6O_2$; $^1$H NMR (600 MHz, CDCl$_3$): δ3.73 (2H, s, H-3), 7.25–7.31 (2H, m, H-4, 6), 7.08–7.14 (2H, m, H-5, 7); $^{13}$C NMR (CDCl$_3$): δ33.07 (C-3), 110.88 (C-7), 123.13 (C-5), 124.18 (C-4), 124.72 (C-9), 128.98 (C-6), 154.82 (C-8), and 174.15 (C=O). Based on the spectrum analysis and comparison with the literature [33,34], compound **1** was identified as 2-coumaranone (Figure 4a).

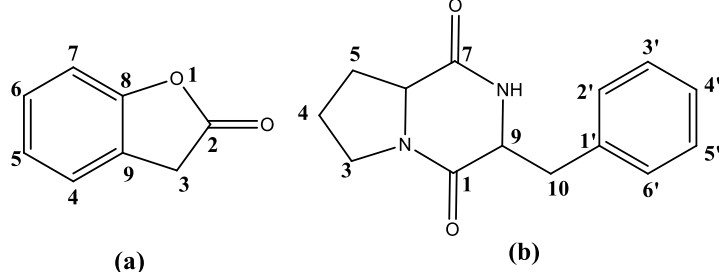

**(a)**          **(b)**

**Figure 4.** Chemical structures of 2-coumaranone (**a**) and cyclo-(Phe-Pro) (**b**).

Compound **2** purified from Fr.5 was a white powder with a relative molecular mass of 224.121, with the chemical formula of $C_{14}H_{16}N_2O_2$. $^1$H NMR (600 MHz, CDCl$_3$): δ7.22–7.34 (5H, m, H-2, 3, 4, 5′, 6′), 4.26 (1H, m, H-6), 4.07 (1H, H-9), 3.65, 3.59 (2H, m, H-3), 3.54, 2.78 (2H, m, H-10), 2,33, 1.88 (each 1H, m, H-5), 2.01(2H, m, H-4). $^{13}$C NMR (CDCl$_3$): δ169.50 (C-7), 165.15 (C-1), 136.01(C-1), 129.38 (C-3′, 5′), 129.20 (C-2′, 6′), 127.66 (C-4′), 59.22 (C-9), 56.28 (C-6), 45.55 (C-3), 36.87 (C-10), 28.44 (C-5), and 22.63 (C-4). According to EIMS and NMR data combined with the related literature, compound **2** was determined to be cyclo-(Pro-Phe) [35,36] (Figure 4b).

### 3.3. Nematicidal Activity against PWN of the Nematicidal Compounds

To determine the nematicidal activity, the semi-lethal concentrations against PWN at 24 h for 2-coumaranone and cyclo-(Pro-Phe) were detected. The LC$_{50}$ values of 2-

coumaranone and cyclo-(Phe-Pro) were 196 and 425 mM, respectively, which indicated that 2-coumaranone had a stronger lethal effect against PWN (Table 3).

**Table 3.** LC$_{50}$ values of 2-coumaranone and cyclo-(Phe-Pro) against PWN for 24 h.

|  | LC$_{50}$ (mM) | 95% Confidence Intervals | Probit Equation |
|---|---|---|---|
| Cyclo-(Pro-Phe) | 0.425 | 0.374–0.483 | y = 2.788x − 5.663 |
| 2-Coumaranone | 0.196 | 0.167–0.233 | y = 2.128x − 3.024 |

*3.4. Effect of the Nematicides on PWN Egg Hatching*

After being treated with 2-coumaranone and cyclo-(Phe-Pro) at concentrations of 0.1, 0.2, and 0.5 mM for 48 h, the hatching of eggs for PWN was significantly inhibited compared to the control ($p < 0.05$) (Figure 5). The hatching rates for eggs treated with 0.1 mM 2-coumaranone and cyclo-(Pro-Phe) were 50.77% and 59.92%, respectively.

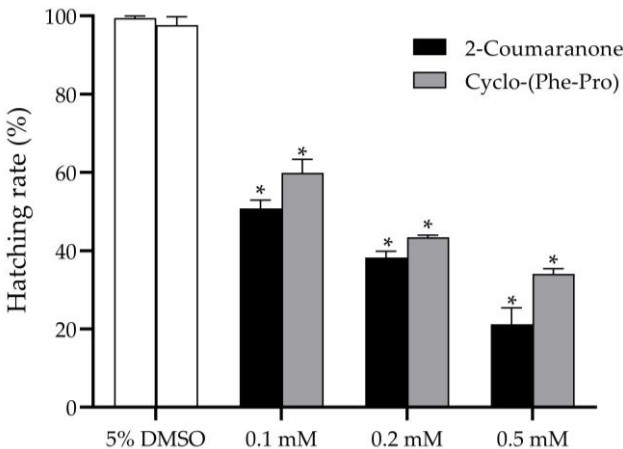

**Figure 5.** Effects of nematicidal compounds on cumulative hatching rates after 48 h. Mean values of four replications were plotted with SD. The asterisk denoted significant difference at $p < 0.05$ compared to controls according to Student's *t*-test.

*3.5. Influence of the Nematicides on PWN Feeding and Population*

To investigate the nematicidal effect on reproduction, PWNs were treated with 0.1 mM 2-coumaranone and cyclo-(Phe-Pro) for 24 h and then inoculated into *B. cinerea*. In contrast to the depleted mycelia in the control groups treated with 5% DMSO and sterile water at the end of 7 d, most of the mycelia was left in the treated groups (Figure 6). Specifically, the little mycelia of *B. cinerea* was fed in the 2-coumaranone-treated group. Therefore, the nematicidal compounds significantly reduced the feeding ability of the nematodes.

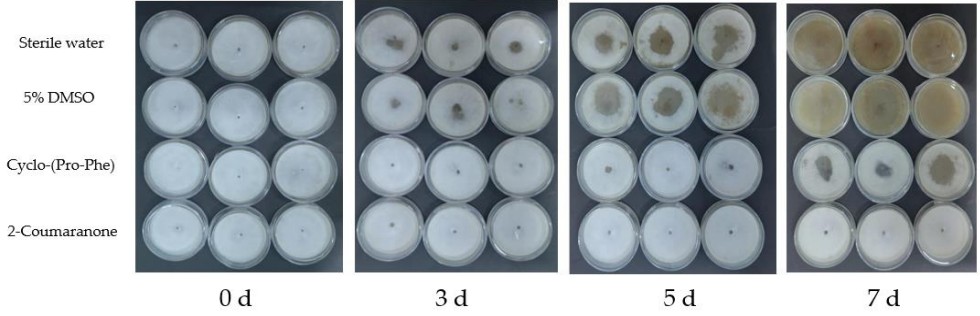

**Figure 6.** The feeding of *Botrytis cinerea* in Petri dishes after inoculation with different PWNs.

Moreover, the nematode populations were 270 and 3400 in the 2-coumaranone and cyclo-(Pro-Phe)-treated groups, respectively, which were significantly lower ($p < 0.01$) than

that of 10,080 and 10,400 in the DMSO and sterile water-treated groups (Figure 7). The inhibition effect on reproduction was significantly higher for 2-coumaranone than that of cyclo-(Pro-Phe) ($p < 0.01$).

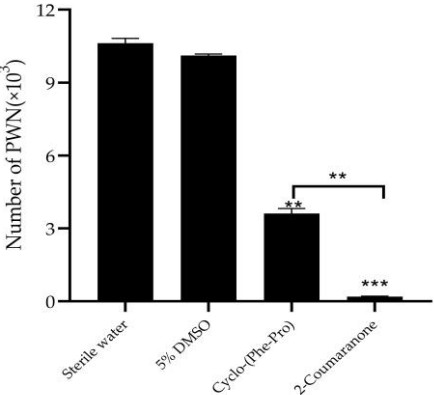

**Figure 7.** Numbers of PWN per dish treated with 2-coumaranone, cyclo-(Phe-Pro), 5% DMSO, and sterile water. Mean values of four replications were plotted with SD. ** and *** represent significant difference at $p < 0.01$ and $p < 0.001$, respectively, compared to control according to Student's *t*-test.

### 3.6. Detection of ROS and Antioxidant Enzymes in PWN

Fluorescence was revealed in the PWNs treated with 2-coumaranone and cyclo-(Pro-Phe) based on DCFH-DA fluorescence staining (Figure 8A–D), which indicated that the two nematicides stimulated the production of ROS in PWN. Furthermore, the activities of CAT and POD in nematodes treated with the nematicidal compounds were significantly lower, while the SOD activity was significantly higher, compared to that in the sterile water and 5% DMSO-treated groups (Figure 9). The results indicate that the accumulation of a large amount of $H_2O_2$ in the nematodes eventually led to their deaths.

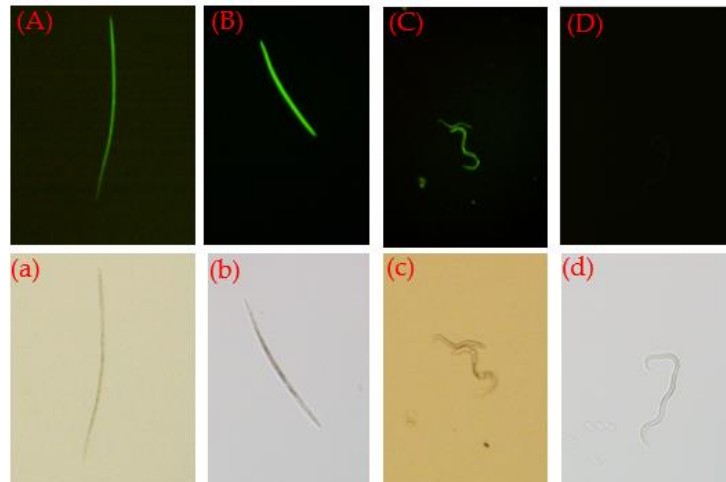

**Figure 8.** Reactive oxygen fluorescence staining of *B. xylophilus*. *B. xylophilus* was imaged in the darkfield (**A–D**) and the corresponding brightfield (**a–d**). The a–d groups were treated with 0.2 mM 2-coumaranone, 0.2 mM cyclo-(Phe-Pro), reactive oxygen hydrogen donor, and 5% DMSO, respectively.

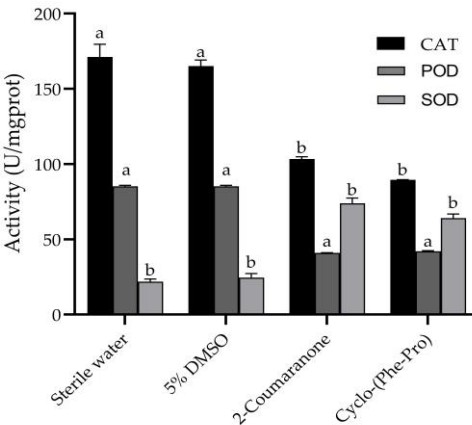

**Figure 9.** Activity of antioxidant enzymes of PWN. Means with the different superscript letters were significantly different at $p < 0.05$ according to LSD test.

*3.7. Nematicidal Activity of the Nematicidal Compounds against PWN In Vivo*

To understand the nematicidal effect of 2-coumaranone and cyclo-(Pro-Phe) in vivo, the PWN-infected Japanese black pine seedlings were treated with the two compounds. On the ninth day after inoculation with PWN, wilting symptoms of seedlings and dehydration of pine needles were observed in the group treated with PWN + 5% DMSO (Figure 10a(A)), while no obvious change was observed in the groups treated with PWN + 2-coumaranone and PWN + cyclo-(Pro-Phe) (Figure 10a(B,C)). On day 12, the pine seedlings treated with PWN + 5% DMSO almost dried up and died (Figure 10b(A)). However, the seedlings treated with PWN + nematicides only showed slight wilting (Figure 10b(B,C)), which indicated that the treatment with 2-coumaranone and cyclo-(Pro-Phe) effectively reduced the wilting symptoms of the Japanese pine seedlings caused by PWN. In contrast with the control group just treated with sterile water (Figure 10a(D),b(D)), no visible side effects were found in seedlings only treated with the nematicides (Figure 10a(E,F),b(E,F)).

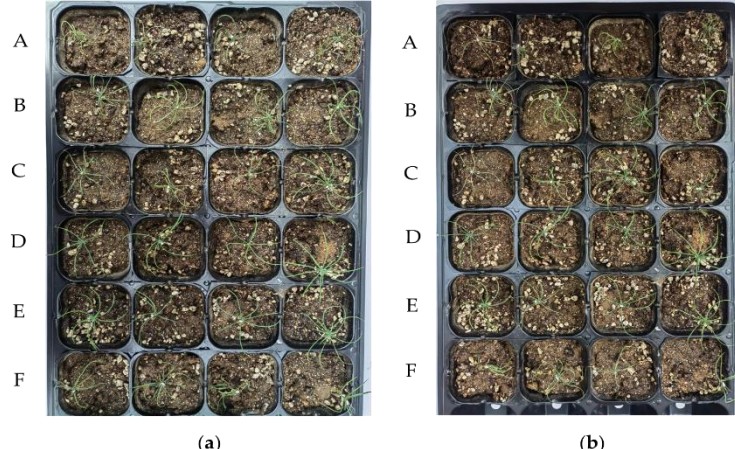

**Figure 10.** Effect of 2-coumaranone and cyclo-(Phe-Pro) on inhibit PWN in vivo. A–F groups were treated with 5% DMSO + PWN, 2-coumaranone + PWN, cyclo-(Pro-Phe) + PWN, 5% DMSO, 50 mM 2-coumaranone, and 50 mM cyclo-(Pro-Phe), respectively. (**a**) Pine seedling symptoms on day 9 in different treatment groups; (**b**) pine seedling symptoms on day 12 in different treatment groups.

## 4. Discussion

Numerous species of microorganisms were tried in the biological control of PWD. Kang et al. [37] screened a nematicidal strain, *Streptomyces* sp. 680,560 among endophytic actinomycetes of Korean pines, and isolated a strongly active compound, teleocidin B4, from its metabolites, which had a significant inhibitory effect on the egg hatching and

development of PWN. *Bacillus thuringiensis* was confirmed to produce endotoxins with strong nematicidal activity against PWN [38,39]. However, few of them were reported to be effective in pine forest tests, and no nematicidal bacteria were reported to be screened from the rhizosphere of pine trees. The microorganisms in the rhizosphere soil of pine trees differ greatly during different periods of infection with PWD, and some of these bacteria might have antagonistic effects against PWD [16]. Therefore, we tried to isolate bacteria with nematicidal activity from healthy Japanese black pine rhizosphere in the current study in order to obtain bioactive compounds as nematicidal candidates against PWN. A nematicidal bacterium, *Lysinimonas* sp M4, was isolated and the antagonistic effect against PWN of this genus was first reported. Several bacterial species were found to have nematicidal activity, among which *Bacillus* genus was the most reported. Some bacteria were confirmed to kill PWN through producing toxins or secreting extracellular proteases to digest nematode body surface tissues, and some others control PWN by interfering with nematode behavior or preying on nematodes [37–42]. We isolated two nematicides against PWN, 2-coumaranone, and cyclo-(Pro-Phe), from the fermentation broth of *Lysinimonas* sp. M4, and several strains of this genus were recorded to be resistant to metal ion stress [43–45], which implies the potential of this genus in PWN control.

The nematicidal compound cyclo-(Phe-Pro), belonging to the cyclic dipeptides (2,5-diketopiperazine), was identified in the present study. Park et al. [35] investigated the nematicidal metabolites of *Bacillus thuringiensis* JCK-1233, and found cyclo-(L-Pro-L-Phe) displayed an inhibitory effect on PWN in pine trees. However, the mechanism of the nematicidal effect of cyclic dipeptides was not further studied in their research. Lee et al. [46] revealed that cyclo-(Phe-Pro) could induce DNA damage in mammalian cells via ROS accumulation, which eventually causes apoptosis. In our research, cyclo-(Phe-Pro), as well as 2-coumaranone, were confirmed to cause the accumulation of ROS in nematodes, which could partially explain the nematicidal mechanism of the two nematicides. Cyclic dipeptides can be produced by bacteria, fungi, plants, and animals in nature [47,48], and have various biological functions, such as being antifungal, antibacterial, antiviral, antitumor, and cell signaling [49,50]. The cyclo-(Phe-Pro) secreted by the bacteria from the rhizosphere of pine trees might help pine trees to inhibit PWN reproduction and relieve pathogenicity symptoms of PWD [51]. Moreover, 2-coumaranone with strong nematicidal activity in the metabolites of *Lysinimonas* sp. M4 has nematicidal activity against PWN, which was not reported. The agent has potential to be developed as a nematicide or a lead compound for a more powerful nematicide by further modification in PWD management on a large scale.

ROS includes superoxide anion ($\cdot O^{2-}$), hydrogen peroxide ($H_2O_2$), and hydroxyl radicals ($\cdot OH$), which play key roles in cell metabolism, oxidative stress, signal transduction, and apoptosis [52]. $H_2O_2$ is mainly produced by SOD, while CAT and POD can remove $H_2O_2$ [53]. Analysis on the aforementioned antioxidant enzymes of the nematicide-treated nematodes in vivo revealed that the activities of CAT and POD were significantly reduced compared to the control, while the activity of SOD was significantly increased. According to the result, we inferred that excessive $H_2O_2$ production might lead to the death of pine nematodes. The levels of other antioxidant enzymes, such as glutathione peroxidases (GPX), are worthy of further study for further elucidating the mechanism of the nematicides isolated from *Lysinimonas* sp. M4 [54].

## 5. Conclusions

In this study, *Lysinimonas* sp. M4 with nematicidal activity was isolated from microorganisms from the rhizosphere soil of Japanese black pine, and two nematicidal compounds, 2-coumaranone and cyclo-(Phe-Pro), were purified from its culture. Both of the nematicidal compounds significantly inhibited egg hatching, feeding, and reproduction of PWN. The two nematicides caused the accumulation of ROS in PWNs, as well as a significant decrease in CAT and POS activity and an increase in SOD activity in PWNs, which might contribute to their nematicidal activity against PWN. Furthermore, both 2-coumaranone

and cyclo-(Phe-Pro) could effectively reduce the wilt symptoms of Japanese black pine seedlings caused by PWN inoculation. This study provides a promising novel bacterial strain and two derived nematicidal compounds for controlling PWD.

**Author Contributions:** Conceptualization, Y.S., Q.G., Q.X., C.W., G.D. and R.L.; methodology, Y.S., C.W., G.D., R.L. and Q.G.; software, Y.S., R.L.; validation, Y.S., R.L., Q.X. and Q.G.; formal analysis, Y.S., G.D. and Q.G.; investigation, Y.S., W.D., H.Y.; resources, C.W., H.Y., G.D. and Q.G.; data curation, W.D., R.L.; writing—original draft preparation, Y.S., G.D., Q.G.; writing—review and editing, R.L., Q.G., W.D., Q.X.; visualization, C.W., G.D., Q.G., W.D.; supervision, Q.X., Q.G. and G.D.; project administration, Q.G. and Q.X.; funding acquisition, Q.G. and R.L. All authors have read and agreed to the published version of the manuscript.

**Funding:** This research was funded by Natural Science Foundation of Shandong Province, grant number ZR2020MC123.

**Institutional Review Board Statement:** Not applicable.

**Informed Consent Statement:** Not applicable.

**Conflicts of Interest:** The authors declare no conflict of interest.

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
