# Peer review of "Two Nematicidal Compounds from Lysinimonas M4 against the Pine Wood Nematode, Bursaphelenchus xylophilus"

_forests, doi:10.3390/f13081191_

Round 1

Reviewer 1 Report

This is rather complex paper dealing with research of nematicidal compounds from soil bacteria. The authors were able to isolate and identified two nematicidal compounds and study its mode of action as well as biological activity using in vivo test. Work is described quite clearly; results are presented in short but understandable form. Considering the Discussion part, I am lacking the more detailed description of utilization of the obtained results; what is the authors’ opinion considering the practical application of tested bacterial isolate (or detected nematicidal compounds) to reduce B. xylophilus infestation of pine trees? What should be investigated in future to secure its utilization?

Minor remarks:

l. 69: probably Baermann funnel

l. 161 and 166: please use the single form of numbers writing (10,000 vs. 10000)

l. 164: add model of microscope

l. 184: unexpected phytotoxic effect

Figure 10 is not very illustrative; it will be betted to provide side photographs of seedlings to see the differences.

Author Response

Dear Reviewer 1,

 Thanks a lot for your helpful comments, and point-by-point answers are summarized as follows:

Q1: Line 69: probably Baermann funnel.

A: Yes, this should be Baermann funnel, and we have corrected it. Thanks.

Q2: Line 161 and 166: please use the single form of numbers writing (10,000 vs. 10000).

A2: Thanks. We have revised ‘10000’ in line 166 (line 168 in the revised version)to ‘10,000).

Q3: Line 164: add model of microscope.

A3: Thanks, we have added the model of microscope.

Q4: unexpected phytotoxic effect

Figure 10 is not very illustrative; it will be betted to provide side photographs of seedlings to see the differences.

A4: For the pine seedlings were generally very little, we tried to illustrate their growth condition from various angels, and found the top view can show the treated seedlings more comprehensively in this experiment, therefore, we only took crane shots. This advice is exactly valuable. Thanks a lot. 

 All the revisions made to the manuscript have been marked up using the ‘Track Changes’ function.

Wish you could consider our positively revised manuscript more worthy of publication.

Reviewer 2 Report

Dear Authors, 

I read your manuscript and I must say that it was pleasant time. In my opinion text is well written and understable, although I am not a chemist, so chemical background was a Little bit tough for me.

I found a few minor lets say "errors"

Line 53: plant rhizospheric (add space) 

In all places where is mentioned: add name, type, model and company of microscope (line 101, 152, 164)

Line 119: In control you written that ethyl acetate extract of NB medium was swrved. What about DMSO which was added to experimentak mixes? 

We do not how you tested normality of distribution (shapiro?). Was the statistic made only with ANOVA or other non-parametric tests? Add information

Please check the figures. Add spaces were needed in their discription of X and Y axes) like in Fig. 3, add soaces after Time (h) and corrected mortality (%)). 

Table 3, Figure 9, line 363- made the name of 2-coumaranone and the other compound standarized through whole manuscript (capital lettter or not), in whole manuscript add spacs after 5% (e.g. line 317, 314, 326)

Fig. 7 I would suggest to chnage the Y axis description and add 'number of individuals' or something like that

Discussion: in line 331: "numerous species of microorganisms... " what about their secondary metabolites? Add some info

Line 352: "via" with italics

Author Response

Dear Reviewer 2,

 Thanks a lot for your helpful comments, and point-by-point answers are summarized as follows:

Q1: Line 53: plant rhizospheric (add space)

A1: Thanks. We have added space between ‘ plant’ and ‘rhizospheric’.

Q2: In all places where is mentioned: add name, type, model and company of microscope (line 101, 152, 164)

A2: Yes, exactly. We have supplemented the necessary information of the microscopes. Thanks.

Q3: Line 119: In control you written that ethyl acetate extract of NB medium was served. What about DMSO which was added to experimental mixes? 

A3: The description in the first version was exactly a little confusing. Thanks for this helpful advice. In fact, ethyl acetate extract of NB medium means the substance which was obtained using ethyl acetate to extract the pure NB medium without inoculation of the baterium M4, and we used the same volume ethyl acetate extract solution of pure NB medium (10 mg/mL, dissolved in DMSO) as the aforementioned ethyl acetate extract solution of baterium M4 cultured in NB medium to serve as the control for testing the nematicidal activity of the secondary metabolite secreted by the bacterial strain M4. In the revised version, we corrected the description.

Q4: We do not how you tested normality of distribution (shapiro?). Was the statistic made only with ANOVA or other non-parametric tests? Add information.

A4: Thanks, we added the relative information to clarify the statistic analysis.

Q5: Please check the figures. Add spaces were needed in their discription of X and Y axes) like in Fig. 3, add spaces after Time (h) and corrected mortality (%)). 

A5: Thanks. We added the spaces in the corresponding place.

Q6: Table 3, Figure 9, line 363- made the name of 2-coumaranone and the other compound standarized through whole manuscript (capital lettter or not), in whole manuscript add spacs after 5% (e.g. line 317, 314, 326)

A6: Yes, exactly. We standarized the initial letter of the names of two compounds. In the revised version, we used lowercase letter in the text, and capital letter in Tables or Figures.

Q7: Fig. 7 I would suggest to change the Y axis description and add 'number of individuals' or something like that.

A7: Thanks for the helpful advice. We have corrected the description of Y axis to ‘number of PWN’.

Q8: Discussion: in line 331: "numerous species of microorganisms... " what about their secondary metabolites? Add some info.

A8: Thanks. We have added some related information with corresponding reference.

Q9: Line 352: "via" with italics.

A9: Thanks. We have italicized the "via".

. All the revisions made to the manuscript have been marked up using the ‘Track Changes’ function.

Wish you could consider our positively revised manuscript more worthy of publication.
